# Trend and determinants of complete vaccination coverage among children aged 12-23 months in Ghana: Analysis of data from the 1998 to 2014 Ghana Demographic and Health Surveys

**Eugene Budu**[1]*, **Eugene Kofuor Maafo Darteh**[1], **Bright Opoku Ahinkorah**[2], **Abdul-Aziz Seidu**[1,3], **Kwamena Sekyi Dickson**[1]

**1** Department of Population and Health, University of Cape Coast, Cape Coast, Ghana, **2** School of Public Health, Faculty of Health, University of Technology Sydney, Sydney, Australia, **3** College of Public Health, Medical and Veterinary Sciences, James Cook University, Townsville, Queensland, Australia

* budueugene@gmail.com

**Data Availability Statement:** The 2014 GDHS reported that ethical approval was granted by the

## Abstract

### Background

Vaccination is proven to be one of the most cost-effective measures adopted to improve the health of children globally. Adhering to vaccines for children has the propensity to prevent about 1.5 million annual child deaths globally. This study sought to assess the trend and determinants of complete vaccination coverage among children aged 12–23 months in Ghana.

### Materials and methods

The study was based on data from four rounds of the Ghana Demographic and Health Survey (GDHS 1998, 2003, 2008, and 2014). Information on 5,119 children aged 12–23 months were extracted from the children's files. Both bivariate and multivariate analyses were conducted to assess the factors associated with complete vaccination and statistical significance was pegged at p<0.05.

### Results

We found that complete vaccination coverage increased from 85.1% in 1998 to 95.2% in 2014. Children whose mothers were in rural areas [aOR = 0.45; CI = 0.33–0.60] had lower odds of getting complete vaccination, compared to those whose mothers were in urban areas. Also, children whose mothers had a secondary level of education [aOR = 1.87; CI = 1.39–2.50] had higher odds of receiving complete vaccination, compared to those whose mothers had no formal education. Children whose mothers were either Traditionalists [aOR = 0.60; CI = 0.42–0.84] or had no religion [aOR = 0.58, CI = 0.43–0.79] had lower odds of receiving complete vaccination, compared to children whose mothers were Christians.

Institutional Review Board of ICF International and Ghana Health Service Ethics Review Committee [12]. Both written and informed consent were obtained from the respondents. We further obtained permission from the DHS Program for use of this data for the study. The dataset is freely available for public use on www.measuredhs.com. Questionnaires used for the survey are appended to the final report published, can be found at: http://dhsprogram.com/publications/publication-FR307-DHS-Final-Reports.cfm

**Funding:** The author(s) received no specific funding for this work.

**Competing interests:** The authors have declared that no competing interests exist.

**Abbreviations:** LMICs, Low and Middle-Income Countries; SSA, Sub-Saharan Africa; GDHS, Ghana Demographic and Health Survey; COR, Crude Odds Ratio; AOR, Adjusted Odds Ratio; CI, Confidence Interval; EPI, Extended Programme on Immunization; WHO, World Health Organization.

## Conclusion

The study revealed that there has been an increase in the coverage of complete vaccination from 1998 to 2014 in Ghana. Mother's place of residence, education, and religious affiliation were significantly associated with full childhood vaccination. Although there was an increase in complete childhood vaccination, it is imperative to improve health education and expand maternal and child health services to rural areas and among women with no formal education to further increase complete vaccination coverage in Ghana.

## Background

Vaccination is a proven cost-effective intervention to reduce childhood morbidity and mortality globally [1, 2]. Due to this and other public health interventions, child health has significantly improved over the years, with child survival rates increasing [3]. Despite these efforts, vaccine-preventable diseases (VPDs) contribute about 17% of the annual under-five mortalities globally [4]. In Ghana, one in eleven children dies before their fifth birthday, largely from preventable childhood diseases [5].

Globally, about three-quarters of the world's children population are reached with the required vaccines, but in sub-Saharan Africa (SSA), a little above half of the children get access to basic vaccination [6], and in poorer remote areas of low- and middle-income countries (LMICs), only 1 in 20 children has access to vaccination [6]. More than 10 million children in LMICs die every year due to lack of access to effective interventions such as vaccination that could fight common and preventable childhood illnesses [7]. This phenomenon resulted in increased global attention which led to the initiation of the Expanded Programme on Immunisation (EPI) in 1974, where less than five percent of children globally were immunised [8].

In Africa, EPI was launched in 1978 and by the mid-eighties, all countries had established National Immunisation Programmes. Ghana, just like other African countries, accepted and launched the EPI in 1978 [9]. The objective of Ghana's EPI was to fully vaccinate 80% of its children aged 0–11 months by 1983 [10]. However, after 20 years of the launch of the programme, the proportion of fully vaccinated children before age one was 51% nationally [11].

According to the 2014 Ghana Demographic and Health Survey (GDHS) report, there has been an increase in vaccination coverage over the years from 47% in 1988 to 79% in 2008. However, there was a decline to 77% in 2014 [12]. There exist urban-rural differences and regional disparities concerning vaccination coverage in Ghana [6]. Some children do not receive any of the vaccines [13].

Recently, some empirical studies have been undertaken regarding vaccination in Ghana. Most of these studies concentrated on issues relating to a particular type of vaccine against a specific health outcome [13–17] or were conducted in some selected places in Ghana [18, 19]. Also, some of the studies conducted have focused on the inequalities in childhood immunisation [20, 21]. To the best of our knowledge, none of these studies has considered the trend and factors associated with complete vaccination coverage among children aged 12–23 months. Therefore, the present study sought to assess the trend and factors associated with childhood vaccination in Ghana, with emphasis on children under 24 months to fill the gap and contribute to the discourse of childhood vaccination. Findings from such a nationwide study have the potential to augment existing mechanisms to scale up vaccination coverage in Ghana to reduce childhood morbidities and mortalities.

## Materials and methods

### Study setting

The Republic of Ghana, the study area, is located on the West African coast, and it has a total land area of 238,533 square kilometres [22]. It is bounded by three French-speaking countries (Burkina Faso on the north, Togo in the east, and Cote d'Ivoire on the west). The south of Ghana lies the 560-kilometres-long Gulf of Guinea. Ghana lies between latitudes 4° and 12°N and longitudes 4°W and 2°E, and the Greenwich Meridian line passes through the sea point of Tema about 24 kilometres to Accra, the capital of the country. Ghana is a low-lying country with a few series of hills and mountains on the eastern border. The population of Ghana, according to the National Population Census conducted in 1960, 1970, 1984, 2000, and 2010 stood at 6,726,815; 8,559,313; 12,296,081; 18,912,079, and 24,658,823 respectively [22]. As at the time of the surveys, the country had 10 regions. These were Western Region, Central Region, Greater Accra Region, Volta Region, Eastern Region, Ashanti Region, Brong Ahafo Region, Northern Region, Upper East Region, and Upper West Region. However, the regions are now 16, namely Oti Region, Brong Ahafo Region, Bono East, Ahafo Region, North East Region, Savannah Region, Western North Region, Western Region, Greater Accra Region, Central Region, Eastern Region, Upper East Region, Upper West Region, Volta Region, Northern Region and Ashanti Region. Ghana has about 51 percent of its population living in urban centres and 49 percent in the rural areas. About 3,405,406, representing 13.8 percent of the total population of the country, are below age 5 [23].

The dominant ethnic groups in Ghana are Akan (47.5%), Mole Dagbani (16.6%), Ga-Adangbe (7.4%), Gruma (5.7%), Guan (3.7%), Grusi (2.5%) with the rest being Mande, Hausa and other ethnic groups. With religious affiliation, the majority of Ghanaians (71.2%) are Christians (Catholic, Protestant, Pentecostal/Charismatic and other Christian) followed by Islam (17.6%), Traditionalist (5.2%), No Religion (5.3%) and 0.8% of the population with Other Religion [22].

### Data source

The study used data from the GDHS 1998, 2003, 2008, and 2014. Specifically, the birth recodes files were used for the study. The GDHS is a nationwide survey that is carried out every five years since it began. The survey gathers information on fertility, family planning, infant and child mortality, maternal and child health, and nutrition. The GDHS focused on child and maternal health and is designed to provide adequate data to monitor the population and health situation in Ghana. The survey adopted a two-stage sampling design. The first stage was characterised by the selection of clusters across urban and rural locations from the entire nation. These made up enumeration areas for the study. The second stage involved the selection of households from the predefined clusters. Details of the methodologies employed in the various rounds of the surveys can be found in the final reports [12] which are also available online at (www.measuredhs.com). In this study, only women (n = 1,135 in 2014, 1,197 in 2008, 1,447 in 2003, and 1,282 in 1998) with children aged 12–23 months were used. After appending all the four datasets, a total of 5,119 respondents were obtained.

### Study variables

**Outcome variable.** The study used complete vaccination as the outcome variable. In this study, complete and full vaccination coverage are used interchangeably. The information on vaccination coverage was collected from either vaccination cards or mothers' verbal responses to these questions: "Did (NAME) ever receive vaccination against Measles?", "Did (NAME)

ever receive vaccination against Polio?", "Did (NAME) ever receive vaccination against BCG?" and "Did (NAME) ever receive vaccination against DPT?". Responses were "Yes", "No" and "Don't Know". These were coded as "No" = 0, "Yes = 1" and "Don't Know = 8". For the analysis, only women who provided definite responses (either "Yes" or "No") were included in the study. According to the WHO guideline (2017), "complete or full vaccination" coverage is defined as a child that has received one dose of BCG, three doses of pentavalent, pneumococcal conjugate (PCV), oral polio vaccines (OPV); two doses of Rotavirus and one dose of measles vaccine. We recoded each question on the specific vaccines as "0 = No" and "1 = Yes" for children who didn't take the recommended doses and those who took, respectively. Complete vaccination was obtained by creating a composite variable which comprised all the vaccines administered. To provide a binary outcome, the two responses (No and Yes) were coded as follows: "Incomplete" = 0, "Complete = 1".

**Independent variable.** The study used nine independent variables that showed statistical significance in previous studies [13–17]. These were age, region, educational level, wealth index, parity, ethnicity, place of residence, religion, and occupation. The age of respondents was categorised into 15–19, 20–24, 25–29, 30–34, 35–39, 40–44, and 45–49. The region was captured as Central, Western, Eastern, Greater Accra, Ashanti, Brong Ahafo, Volta, Northern, Upper East, and Upper West. Educational level was classified into no education, primary, secondary, and higher. Place of residence was coded as rural and urban. Marital status was recoded into married, never married, widowed, divorced, and cohabitating. Wealth status was grouped into poorest, poorer, middle, richer, and richest. Parity was coded from a question that assessed the number of children a woman had ever given birth to. Responses were grouped into no birth (that is, before current pregnancy), one birth, two births, three births, and four or more births. The occupation was recoded into working and not working. Also, ethnicity was recoded into Akan (which consists of Asante, Akwapim, Fante, and others), Ga-Adangbe, Ewe, Guan, Mole-Dagbani, Grussi, Gruma, Mande and Others. Religion was recoded into Christian, Islam, Traditional, and no religion.

## Statistical analyses

Data were processed and analysed using STATA version 14.0, with the use of both inferential and descriptive statistics. The analysis was done in three steps. First, a graphical representation of the proportion of complete and incomplete vaccination from 1998 to 2014 was done (see Fig 1). In the second and third steps of the analysis, 3 of the data points (that is, 1998, 2003, 2008) were appended to the 2014 dataset. In the second step, descriptive statistics (frequency and percentages) were used to describe the characteristics of the respondents (see Table 1), and cross-tabulation of all the independent variables against the outcome variable was done using chi-square test of independence. All the variables that showed statistical significance (p<0.05) were moved to the multivariable level. The third step was the multivariable analysis, using a binary logistic regression. At the multivariable analysis, a binary logistic regression model was fitted. The results were presented as crude odds ratios (cORs) and adjusted odds ratios (aORs), with their corresponding 95% confidence intervals (CIs) signifying their level of precision. Statistical significance was declared at p<0.05. Sample weight (v005/1,000,000) was applied and the survey command (svy) was used to account for the complex sampling design of the survey.

## Ethical approval

The 2014 GDHS reported that ethical approval was granted by the Institutional Review Board of ICF International and Ghana Health Service Ethics Review Committee [12]. Both written

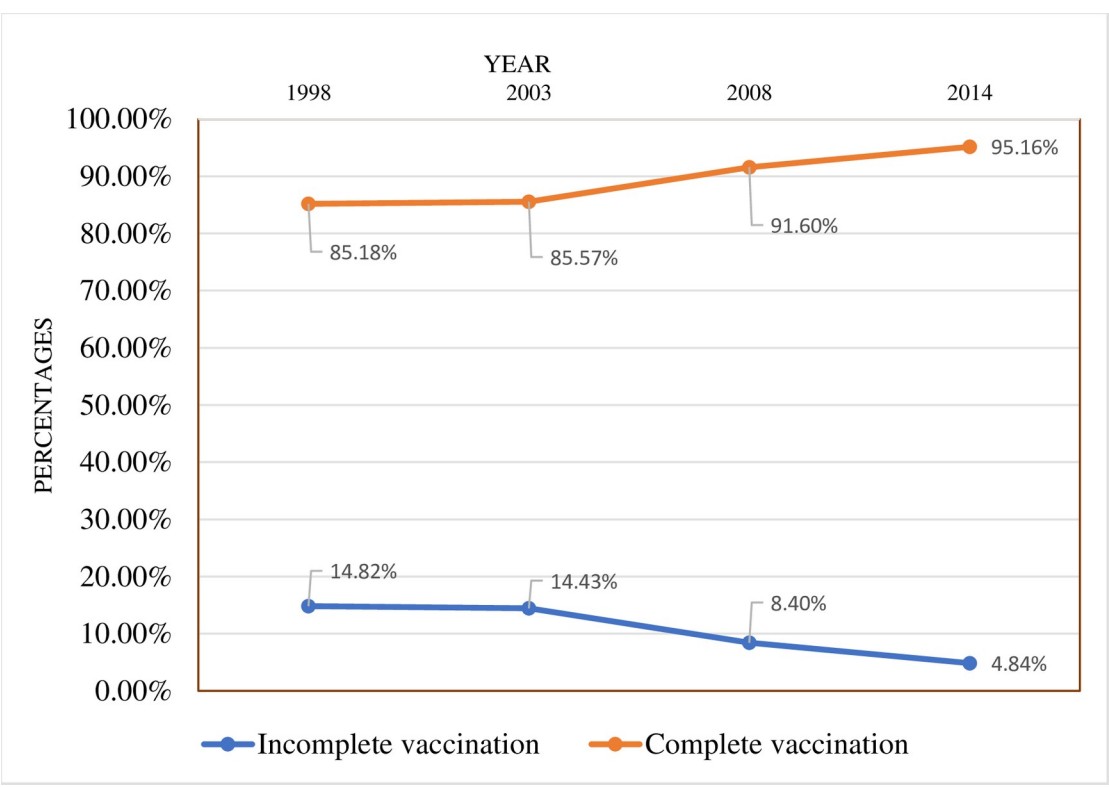

**Fig 1. Vaccination coverage from 1998 to 2014.**

and informed consent was obtained from the respondents. We further obtained permission from the DHS Program for use of the data for this study. The dataset is freely available for public use on www.measuredhs.com. Questionnaires used for the survey are appended to the final report published and can be found at http://dhsprogram.com/publications/publication-FR307-DHS-Final -Reports.cfm.

# Results

## Descriptive results

**Coverage of childhood vaccination from 1998–2014.** Fig 1 is a graphical representation of the childhood vaccination coverage from 1998 to 2014.

The figure shows that complete vaccination coverage has increased from 85.1% in 1998 to 95.2% in 2014 while incomplete vaccination has reduced from 14.9% to 4.8% (Fig 1).

## Socio-demographic characteristics of women and full vaccination coverage for their children

Table 1 presents the results on socio-demographic characteristics and the proportion of childhood vaccination across socio-demographic characteristics. It was found that 26.9% of the respondents were aged 25–29 years. More than 6 in 10 were from rural areas. Forty-one percent of the respondents had no formal education while 75.3% of the respondents were married. More than 1 in 10 of the respondents were mothers who lived in the Northern Region. The majority of the respondents (84.4%) were mothers who were working whereas 31.3% were mothers who belonged to the Akan ethnic group. About 42% of the respondents were mothers

**Table 1. Sociodemographic characteristics of women with children 12–23 months of age.**

| Variables | Weighted N | Weighted % | Complete vaccination | χ2/p-value |
|---|---|---|---|---|
| **Age** | | | | 10.3/0.112 |
| 15–19 | 319 | 6.2 | 87.2 | |
| 20–24 | 1,163 | 22.7 | 88.7 | |
| 25–29 | 1,377 | 26.9 | 89.4 | |
| 30–34 | 1,043 | 20.4 | 91.2 | |
| 35–39 | 784 | 15.3 | 88.7 | |
| 40–44 | 332 | 6.5 | 87.1 | |
| 45–49 | 101 | 2.0 | 84.2 | |
| **Residence** | | | | 98.0/<0.001 |
| Urban | 1,577 | 30.8 | 95.6 | |
| Rural | 3,542 | 69.2 | 86.2 | |
| **Educational level** | | | | 102.0/<0.001 |
| No education | 2,107 | 41.2 | 84.3 | |
| Primary | 1,068 | 20.9 | 89.2 | |
| Secondary | 1,838 | 35.9 | 94.0 | |
| Higher | 106 | 2.1 | 97.2 | |
| **Marital status** | | | | 8.9/0.063 |
| Never married | 229 | 4.5 | 92.1 | |
| Married | 3,855 | 75.3 | 89.0 | |
| Cohabiting | 793 | 15.5 | 89.4 | |
| Widowed | 42 | 0.8 | 97.6 | |
| Divorced | 200 | 3.9 | 85.0 | |
| **Region** | | | | 106.1/<0.001 |
| Western | 505 | 9.9 | 89.7 | |
| Central | 426 | 8.3 | 90.6 | |
| Greater Accra | 441 | 8.6 | 95.2 | |
| Volta | 428 | 8.4 | 86.5 | |
| Eastern | 475 | 9.3 | 89.7 | |
| Ashanti | 660 | 12.9 | 91.2 | |
| Brong Ahafo | 484 | 9.5 | 89.7 | |
| Northern | 739 | 14.4 | 79.3 | |
| Upper East | 472 | 9.2 | 92.0 | |
| Upper West | 472 | 9.6 | 92.0 | |
| **Occupation** | | | | 0.4/0.535 |
| Not working | 798 | 15.6 | 88.5 | |
| Working | 4,321 | 84.4 | 89.2 | |
| **Ethnicity** | | | | 112.8/<0.001 |
| Akan | 1,601 | 31.3 | 92.4 | |
| Ga-Adangbe | 239 | 4.7 | 94.1 | |
| Ewe | 611 | 11.9 | 90.2 | |
| Guan | 298 | 5.8 | 85.6 | |
| Mole-Dagbani | 1,094 | 21.4 | 90.5 | |
| Grussi | 323 | 6.3 | 88.2 | |
| Gruma | 292 | 5.7 | 76.7 | |
| Mande | 179 | 3.5 | 91.1 | |
| Others | 482 | 9.4 | 80.7 | |
| **Parity** | | | | 14.4/0.002 |

(*Continued*)

**Table 1.** (Continued)

| Variables | Weighted N | Weighted % | Complete vaccination | χ2/p-value |
|---|---|---|---|---|
| One birth | 1,059 | 20.7 | 90.8 | |
| Two births | 1,074 | 21.0 | 90.7 | |
| Three births | 857 | 16.7 | 89.9 | |
| Four or more births | 2,129 | 41.6 | 87.2 | |
| **Wealth quintile** | | | | 25.6/<0.001 |
| Poorest | 1,411 | 27.6 | 86.0 | |
| Poorer | 1,077 | 21.0 | 88.8 | |
| Middle | 1,048 | 20.5 | 89.7 | |
| Richer | 941 | 18.4 | 91.1 | |
| Richest | 642 | 12.5 | 92.5 | |
| **Religion** | | | | 92.1/<0.001 |
| Christianity | 3,468 | 67.8 | 91.3 | |
| Islam | 966 | 18.9 | 88.5 | |
| Traditionalist | 335 | 6.5 | 78.2 | |
| No religion | 350 | 6.8 | 79.4 | |
| **Total** | **5,119** | **100.0** | **89.1** | |

Source: 1998–2014 Ghana Demographic and Health Survey, χ2 = Chi-square.

who had had four or more births whereas 27.6% of the respondents belonged in the poorest wealth quintile. About 68 percent of the mothers were Christians.

The results further showed that 91.2% of the respondents whose children had complete vaccination were aged 30–34. About ninety-six percent (95.6%) of the respondents whose children had complete vaccination was in the urban areas whereas 94% had mothers who had secondary education. Moreover, 97.6% of the respondents whose children had complete vaccination were widowed. Also, 95.2% of children who had complete vaccination were children of mothers from the Greater Accra Region while the majority of them (89.2%) had mothers who were working. About 94.1% of the children who received complete vaccination were from mothers who belonged to the Ga-Adangbe ethnic group. Close to 91% of the mothers whose children had complete vaccination have had one birth. Majority of the children (92.5%) who received complete vaccination were with mothers who were in the richest wealth quintile while 91.3% of the children had Christian mothers (see Table 1).

With the chi-square analysis, place of residence (χ2 = 98.0, p<0.001), level of education (χ2 = 102.0, p<0.001), region of residence (χ2 = 106.1, p<0.001), parity (χ2 = 14.4, p<0.002), ethnicity (χ2 = 112.8, p<0.001), wealth quintile (χ2 = 25.6, p<0.001), and religion (χ2 = 92.1, p<0.001) showed a statistically significant association with full childhood vaccination (Table 1).

## Multivariable analysis results of full vaccination coverage among 12–23 months' children in Ghana

The analysis revealed that children whose mothers were in the rural areas [aOR = 0.45; CI = 0.33–0.60] had lower odds of getting complete vaccination, compared to children whose mothers are in urban areas. Also, children whose mothers had secondary level of education [aOR = 1.87; CI = 1.39–2.50] had higher odds of receiving complete vaccination, compared to children whose mothers had no formal education. Children whose mothers were in the Upper East [aOR = 2.24; CI = 1.33–3.77] and Upper West Regions [aOR = 2.16; CI = 1.28–3.65] had

higher odds of receiving complete vaccination, compared to children whose mothers were in the Western Region. Children whose mothers were Traditionalists [aOR = 0.60; CI = 0.42–0.84] or had no religion [aOR = 0.58, CI = 0.43–0.79] had lower odds of receiving complete vaccination, compared to children whose mothers were Christians. Controlling for the years the surveys were conducted, we found that children born in 2008 [aOR = 1.86; CI = 1.35–2.57] and 2014 [aOR = 3.29; CI = 2.25–4.82] survey years had higher odds of receiving complete vaccination, compared to those in the 1998 survey year (Table 2).

**Table 2. Binary logistic regression analysis on complete vaccination among children age 12–23 months.**

| Variables | cOR(95% CI) | aOR (95% CI) |
|---|---|---|
| **Residence** | | |
| Urban | 1 | 1 |
| Rural | 0.29*** (0.23–0.38) | 0.45*** (0.33–0.60) |
| **Educational level** | | |
| No education | 1 | 1 |
| Primary | 1.54*** (1.23–1.93) | 1.13 (0.88–1.47) |
| Secondary | 2.92*** (2.33–3.66) | 1.87*** (1.39–2.50) |
| Higher | 6.38** (2.01–20.22) | 2.15 (0.63–7.37) |
| **Wealth Quintile** | | |
| Poorest | 1 | 1 |
| Poorer | 1.28* (1.01–1.63) | 1.08 (0.82–1.42) |
| Middle | 1.41** (1.10–1.81) | 1.09 (0.81–1.47) |
| Richer | 1.66*** (1.26–2.17) | 1.21 (0.86–1.70) |
| richest | 2.01*** (1.44–2.79) | 1.09 (0.73–1.64) |
| **Region** | | |
| Western | 1 | 1 |
| Central | 1.11 (0.72–1.71) | 0.96 (0.62–1.54) |
| Greater Accra | 2.30** (1.36–3.88) | 1.20 (0.66–2.18) |
| Volta | 0.73 (0.49–1.09) | 0.83 (0.52–1.33) |
| Eastern | 1.00 (0.66–1.51) | 0.83 (0.531.30) |
| Ashanti | 1.19 (0.80–1.77) | 1.05 (0.69–1.62) |
| Brong Ahafo | 1.00 (0.66–1.50) | 0.97 (0.63–1.51) |
| Northern | 0.44*** (0.31–0.62) | 0.69 (0.44–1.07) |
| Upper East | 1.31 (0.85–2.03) | 2.24** (1.33–3.77) |
| Upper West | 1.33 (0.86–2.05) | 2.16** (1.28–3.65) |
| **Ethnicity** | | |
| Akan | 1 | 1 |
| Ga-Adangbe | 1.33 (0.75–2.35) | 1.63 (0.88–3.04) |
| Ewe | 0.76 (0.55–1.05) | 1.04 (0.711.54) |
| Guan | 0.49*** (0.34–0.71) | 0.95 (0.611.46) |
| Mole-Dagbani | 0.79 (0.60–1.03) | 1.04 (0.70–1.56) |
| Grussi | 0.62* (0.42–0.91) | 0.89 (0.55–1.45) |
| Gruma | 0.27*** (0.20–0.38) | 0.67 (0.43–1.06) |
| Mande | 0.84 (0.49–1.45) | 1.85 (0.92–3.69) |
| Others | 0.35*** (0.26–0.46) | 0.72 (0.461.13) |
| **Parity** | | |
| One birth | 1 | 1 |
| Two births | 0.99 (0.74–1.33) | 1.07 (0.79–1.45) |

(*Continued*)

**Table 2.** (Continued)

| Variables | cOR(95% CI) | aOR (95% CI) |
|---|---|---|
| Three births | 0.90 (0.67–1.22) | 1.07 (0.78–1.47) |
| Four or more births | 0.69** (0.54–0.89) | 1.02 (0.78–1.33) |
| **Religion** | | |
| Christianity | 1 | 1 |
| Islam | 0.74** (0.58–0.93) | 0.92 (0.67–1.22) |
| Traditionalist | 0.34*** (0.26–0.46) | 0.60** (0.42–0.84) |
| No religion | 0.37*** (0.28–0.49) | 0.58*** (0.43–0.79) |
| **Survey wave (Year)** | | |
| 1998 | 1 | 1 |
| 2003 | 1.03 (0.83–1.28) | 1.12 (0.84–1.49) |
| 2008 | 1.89*** (1.46–2.44) | 1.86*** (1.35–2.57) |
| 2014 | 3.41*** (2.51–4.62) | 3.29*** (2.25–4.82) |
| **N** | **5,119** | **5,119** |
| **Pseudo R$^2$** | | **0.101** |

cOR = crude Odds Ratio, aOR = adjusted Odds Ratio, 95% confidence intervals in brackets

* $p < 0.05$

** $p < 0.01$

*** $p < 0.001$; 1 = reference category.

Source: 1998–2014 Ghana Demographic and Health Surveys.

## Discussion

Vaccination has been useful in delaying over two million deaths each year globally [4], and it is one of the three most successful public health initiatives [24, 25]. We aimed at assessing the trend and determinants of complete vaccination coverage among children in Ghana. It was seen from the findings of the study that complete vaccination coverage of children has increased from 1998 to 2014. That is, there was an increase of complete vaccination from 85.1% in 1998 to 95.2% in 2014 while incomplete vaccination decreased. This finding is consistent with earlier studies that have observed increasing trends in complete vaccination coverage [26, 28]. This increase could be as a result of the education of women on the benefits of availing their children for vaccination, and urbanisation.

The findings of the study indicate that several maternal factors play a role in the uptake of complete vaccination among children in Ghana. The findings revealed that children born of women in rural areas had lower odds of getting complete vaccination, compared to children whose mothers are in urban areas. Urban areas are argued to have more health facilities and health care professionals than rural areas [27], and this could explain why the highest proportion of children who received complete vaccination had mothers in the urban areas. This finding supports studies by Ushie, Fayehun, and Ugal [26] and Tamirat and Sisay [2], who observed children in urban areas have consistently higher complete vaccination rates than those in rural areas. Also, a study which was conducted in 2014 in Ethiopia revealed that place of residence was an important determinant of full child vaccination in Ethiopia. Relatedly the results of the study also revealed that region, where the mothers of the children were found, was a significant factor that affects the complete vaccination coverage of the child. This finding is in line with studies conducted in Zimbabwe [28] and Ethiopia [29] among children aged 12–23 months.

Results from our study also revealed that mothers' educational level is a significant factor that influences the uptake of complete vaccination among children in Ghana. Specifically, we found that children whose mothers have attained a secondary level of education had higher odds of receiving complete vaccination, compared to children whose mothers have no formal education. This result confirms that of Tamirat and Sisay [2], Hu et. al. [30], and Kamau and Esimai [31] on the association between educational level and complete vaccination coverage. The possible explanation for this finding is that women with a secondary level of education may have adequate knowledge and information about vaccination and child welfare and may avail their children to complete vaccination. They are also more likely to comprehend the health education and the benefits of achieving full vaccination coverage that healthcare providers give during child welfare clinics, compared to those with no formal education.

As observed in previous studies in Ghana [32], Ethiopia [29], and 15 countries in SSA [33], the religious affiliation of women showed statistically significant association with complete vaccination coverage of their children. In particular, children whose mothers are Traditionalists and those whose mothers had no religious affiliation had lower odds of receiving complete vaccination, compared to children whose mothers are Christians. According to Costa et al. [33], greater involvement of religious leaders in vaccine promotion has proven to be effective and may constitute an important strategy for reaching parents whose children are not vaccinated or have not received complete vaccination. We propose further qualitative research to explore the nuances surrounding the religious affiliation of mothers and complete vaccination coverage for their children.

## Strengths and limitations

Due to the cross-sectional design, our results lend itself to association between maternal factors and complete vaccination status of children in Ghana. Causal inference interpretations, however, cannot be made. Notwithstanding, the study has some outstanding strengths worth acknowledging. To the best of our knowledge, this is the first study to assess the trends and determinants of vaccination among children with a national coverage in Ghana. The representativeness, scientifically sound methodological approach, and rigour of the analytical approach make it feasible for generalisation of the findings to other women and their children in Ghana.

## Conclusion

Our study revealed that there has been an increase in the coverage of complete vaccination coverage from 1998 to 2014. Full childhood vaccination was predicated by maternal factors such as mother's place of residence, educational level, and religious affiliation. To further improve and sustain the coverage of full childhood vaccination, interventions targeting improvement in full childhood vaccination should focus on addressing these maternal factors. Also, the Ghana Health Service and the Ministry of Health should strengthen their existing interventions and social and behavioural change communication strategies as a way of improving on full childhood vaccination coverage.

## Acknowledgments

We are grateful to Measure DHS for making thee dataset freely available for our use. We also acknowledge Mr. Ebenezer Agbaglo of the Department of English, University of Cape Coast, who thoroughly copy-edited this manuscript for language usage, spelling and grammar.

## Author Contributions

**Conceptualization:** Eugene Budu.

**Data curation:** Eugene Budu, Abdul-Aziz Seidu.

**Formal analysis:** Eugene Budu, Bright Opoku Ahinkorah, Abdul-Aziz Seidu.

**Funding acquisition:** Eugene Budu, Abdul-Aziz Seidu.

**Investigation:** Eugene Budu, Bright Opoku Ahinkorah, Abdul-Aziz Seidu.

**Methodology:** Eugene Budu, Bright Opoku Ahinkorah, Abdul-Aziz Seidu.

**Project administration:** Eugene Budu.

**Resources:** Eugene Budu, Bright Opoku Ahinkorah, Abdul-Aziz Seidu.

**Software:** Eugene Budu, Abdul-Aziz Seidu.

**Supervision:** Eugene Budu, Eugene Kofuor Maafo Darteh.

**Validation:** Eugene Budu, Bright Opoku Ahinkorah, Abdul-Aziz Seidu.

**Visualization:** Eugene Budu, Abdul-Aziz Seidu.

**Writing – original draft:** Eugene Budu, Eugene Kofuor Maafo Darteh, Bright Opoku Ahinkorah, Abdul-Aziz Seidu, Kwamena Sekyi Dickson.

**Writing – review & editing:** Eugene Budu, Eugene Kofuor Maafo Darteh, Bright Opoku Ahinkorah, Abdul-Aziz Seidu, Kwamena Sekyi Dickson.

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
