## [Decision Letter · Decision Letter 0]

21 May 2020

PONE-D-20-10150

Trends and determinants of complete immunization coverage among Children aged 12-23 months in Ghana: Analysis of data from the 1998 to 2014 Ghana Demographic and Health Surveys

PLOS ONE

Dear Mr Budu,

Thank you for submitting your manuscript to PLOS ONE. After careful consideration, we feel that it has merit but does not fully meet PLOS ONE’s publication criteria as it currently stands. Therefore, we invite you to submit a revised version of the manuscript that addresses the points raised during the review process.

We would appreciate receiving your revised manuscript by Jul 05 2020 11:59PM. To enhance the reproducibility of your results, we recommend that if applicable you deposit your laboratory protocols in protocols.io, where a protocol can be assigned its own identifier (DOI) such that it can be cited independently in the future. For instructions see: http://journals.plos.org/plosone/s/submission-guidelines#loc-laboratory-protocols

We look forward to receiving your revised manuscript.

Kind regards,

Yacob Zereyesus, Ph.D.

Academic Editor

PLOS ONE

Journal Requirements:

2. Please provide additional details regarding participant consent. In the ethics statement in the Methods and online submission information, please ensure that you have specified (1) whether consent was informed and (2) what type you obtained (for instance, written or verbal, and if verbal, how it was documented and witnessed). If the need for consent was waived by the ethics committee, please include this information.

Additional Editor Comments (if provided):

Please see comments below.

Reviewers' comments:

Reviewer's Responses to Questions

**Comments to the Author**

1. Is the manuscript technically sound, and do the data support the conclusions?

Reviewer #1: Yes

2. Has the statistical analysis been performed appropriately and rigorously? 

Reviewer #1: No

3. Have the authors made all data underlying the findings in their manuscript fully available?

Reviewer #1: Yes

4. Is the manuscript presented in an intelligible fashion and written in standard English?

Reviewer #1: Yes

5. Review Comments to the Author

Reviewer #1: Title: Trends and determinants of complete immunization coverage among children aged 12-23 months in Ghana: Analysis of data from the 1998 to 2014 Ghana Demographic and Health Surveys

Comments to the Authors:

General Comments:

Immunization is essential for better health outcomes, particularly, in children. This topic is very relevant to many countries and global health. Therefore, this paper contributes to the discussion on vaccinations and overall acceptance. However, the inconsistent spelling of some words (e.g. immunization vs immunization or trend vs trends), unclear use of “immunization” or “vaccination”, discrepancies between the present results on full immunization (85.1% to 95.2%) and the 2008 GDHS or 2014 GDHS fully immunized coverage (e.g.79% or 77% respectively). These should be revised or explained to improve the quality of the manuscript. In addition, the language standard should be improved. Thus, major revisions are necessary.

Title:

The title is very clear. However, the word “trends” could be changed to “trend” since there is figure 1 which depicts fully immunization.

Abstract:

1. The title used the word “immunization” while the first sentence under the background of the abstract is “vaccination”. The authors should consider this rephrasing either title on the first sentence under the background.

2. On the title, the word “trends” is used while “trend” is applied under background.

3. Inconsistency in the use of “immunization” vs “immunisation” and “full childhood vaccination” vs “full immunization coverage” and “worldwide” vs “globally”. The authors should be consistent.

4. The sentence “Both bivariate and multivariate analysis were….” The word “analysis” should be changed to “analyses”.

5. The logistic results (Odds Ratio and 95%CI) are inaccurate and not found in table 2. Moreover, the table2 has adjusted Odds Ratio (aOR). It is unclear whether the results under the abstract are from crude Odds Ratio (OR).

6. The sentence “The study revealed that there has been an increase in the coverage of complete immunization …. with incomplete vaccination declining”. The second part of the sentence should be removed.

7. From 1998 to 2014, there was about 10.0% improvement in full immunization (85.1% to 95.2%), how did the authors determine “substantial gains”?

8. The “Ethical Review Committee of Ghana Health Service” should be changed to the “Ghana Health Service Ethics Review Committee” often written as GHSERC.

Main Manuscript:

Introduction:

1. The sentence “Adhering to these vaccines for children….. deaths among children globally”. Repeated use of the same word in one sentence affects the quality. The authors should revise this sentence and similar sentences (e.g. “… led to increased globally attention which led to…” to improve the quality of the paper.

2. There is the need for consistency in the use of percentage (e.g. “From 47% in 1988 to 79 percent…” and “.. in 1974 where less than five percent…”

3. The sentences “There was, however, …” and “The present study, therefore,…”should be revised.

Methods:

1. The study site or setting is missing in the methods section. This should be included.

2. The authors should be consistent in the use of terms: “complete vaccination” vs “full immunization coverage” vs “full immunization” vs “full childhood vaccination”,

3. The in-text citation varies between numbers (e.g. [2,3]) and names (Armah et al., 2016). This should be revised appropriately.

4. This sentence is unclear “At the multivariate analysis, a Binary Logistic Regression Model was fitted”. The use of capital letters and clarity of the sentence should be review.

5. In addition, the authors did not show the crude odds ratio before presenting their adjusted logistic regression results. This may be confusing to the readers.

6. The institution that approved is the “Ghana Health Service Ethics Review Committee” not Ethical Review Committee of Ghana Health Service”.

Results:

1. The sentence “From the Figure, it was seen that complete vaccination has increased over the years while income vaccination has been reducing”. This needs revision.

2. The authors have used the word “respondents” inappropriately in the description of the results. (e.g. “More than half of respondents (66.2%) had mothers from rural areas”). This implies that children (12-23 months of age) were the respondents. Rather the mothers were the respondents in the current study. Thus, the entire description of table 1 should be revised.

3. There are many inaccuracies between the description of the results and figures found on tables 1 (e.g. 66.2% instead of 69.2%)

4. Another example, “More than 42 percent instead of 41.2%”. Thus, the presentation of the results under table 1 should be carefully done.

5. The authors also started a couple of sentences with percentages. E.g. 75.2% of the respondents ….., 13.2% of children …, 32.4% of the respondents…”. The authors should consider reviewing the results section of the paper.

6. The authors presented some p-values as “P<0.000”. They should revise it to p<0.001.

7. Both the presentation of the multivariate results in table 2 and the description of the same findings need improvement. The quality of the table is low.

Discussion:

1. Overall, the discussion is short and unclear.

2. The sentence “We aimed at assessing the trends and determinates ….”. The word “determinates” should be revised

3. The sentences “It was seen from the findings…from 1998 to 2014” and “That is, there was increase from 85.1% in 1998 to 95.2% in 2014”. This should be revised.

4. The sentence “…areas had a low odd of getting vaccination…” should also be revised.

5. There are problems with the in-text citation under the discussion. This needs attention and revision.

6. The sentence “This finding confirms the results of a study conducted by [27, 30]”. This sentence is incomplete.

Limitations:

The authors should limit the study limitations to their study. For example, “There is the possibility of social desirability bias…” This may not be appropriate for the present study based on the methods.

Abbreviations:

1. Some abbreviations were not defined at first use e.g. SNNPRS.

References:

1. The references need formatting. For example, it is unclear to know the author of the first reference.

2. The same author is presented in three different formats e.g. World Health Organization, WHO and W.H.O. The authors should use one format.

3. There are also different in-text citations in the paper. The authors should address the inconsistencies.

6. PLOS authors have the option to publish the peer review history of their article (what does this mean?). If published, this will include your full peer review and any attached files.

Reviewer #1: Yes: Martin Nyaaba Adokiya

---

## [Author Response · Author response to Decision Letter 0]

10 Jun 2020

AUTHORS’ RESPONSE TO REVIEWS

Title: Trend and determinants of complete vaccination coverage among children aged 12-23 months in Ghana: Analysis of data from the 1998 to 2014 Ghana Demographic and Health Surveys

Dear Editor and Reviewer (s),

On behalf of all authors, I convey our gratitude to you for the critical and constructive review that has led to the improvement of our paper entitled “Trend and determinants of complete vaccination coverage among children aged 12-23 months in Ghana: Analysis of data from the 1998 to 2014 Ghana Demographic and Health Surveys”. We have revised the manuscript based on the comments raised. In the following detailed response, we address each comment calling for changes point-by-point, indicating where relevant additional texts have been added to the body of the manuscript. Most of the changes have been indicated in yellow colour. We believe the manuscript has improved substantively and will be published in your reputable journal, PLOS ONE 

Version:1 

Manuscript ID: PONE-D-20-10150

Date: 9/06/2020

Reviewer #1: Title: Trend and determinants of complete immunization coverage among children aged 12-23 months in Ghana: Analysis of data from the 1998 to 2014 Ghana Demographic and Health Surveys

Comments to the Authors:

General Comments:

1. Comment: Immunization is essential for better health outcomes, particularly, in children. This topic is very relevant to many countries and global health. Therefore, this paper contributes to the discussion on vaccinations and overall acceptance. However, the inconsistent spelling of some words (e.g. immunization vs immunization or trend vs trends), unclear use of “immunization” or “vaccination”, discrepancies between the present results on full immunization (85.1% to 95.2%) and the 2008 GDHS or 2014 GDHS fully immunized coverage (e.g.79% or 77% respectively). These should be revised or explained to improve the quality of the manuscript. In addition, the language standard should be improved. Thus, major revisions are necessary.

Response: Thank you for the comments. We have revised the manuscript to take care of all the inconsistencies. 

Title

2. Comment: The title is very clear. However, the word “trends” could be changed to “trend” since there is figure 1 which depicts fully immunization.

Response: The authors have changed the word “trends” to “trend” in the title. 

Abstract:

3. Comment: 1. The title used the word “immunization” while the first sentence under the background of the abstract is “vaccination”. The authors should consider this rephrasing either title on the first sentence under the background.

Response: The authors have addressed this comment by rephrasing the title (see page 1)

4. Comment: 2. On the title, the word “trends” is used while “trend” is applied under background.

Response: The authors have addressed this inconsistency.

5. Comment: 3. Inconsistency in the use of “immunization” vs “immunisation” and “full childhood vaccination” vs “full immunization coverage” and “worldwide” vs “globally”. The authors should be consistent.

Response: The authors have addressed the inconsistencies in the entire manuscript. 

6. Comment: 4. The sentence “Both bivariate and multivariate analysis were….” The word “analysis” should be changed to “analyses”.

Response: The authors have changed the world “analysis” to “analyses” [see page 1]

7. Comment: 5. The logistic results (Odds Ratio and 95%CI) are inaccurate and not found in table 2. Moreover, the table2 has adjusted Odds Ratio (aOR). It is unclear whether the results under the abstract are from crude Odds Ratio (OR).

Response: We have corrected the logistic results [see page 1]. Also, the results from the abstract are adjusted odds ratio and not crude odds ratio. 

8. Comment: 6. The sentence “The study revealed that there has been an increase in the coverage of complete immunization …. with incomplete vaccination declining”. The second part of the sentence should be removed.

Response: The authors second part of the sentence has been removed.

9. Comment: 7. From 1998 to 2014, there was about 10.0% improvement in full immunization (85.1% to 95.2%), how did the authors determine “substantial gains”?

Response: The authors have addressed and revised the statement. 

10. Comment: 8. The “Ethical Review Committee of Ghana Health Service” should be changed to the “Ghana Health Service Ethics Review Committee” often written as GHSERC.

Response: The authors have addressed this comment by changing “Ethical Review Committee of Ghana Health Service” to “Ghana Health Service Ethics Review Committee (GHS-ERC)”(see page 5, line 186).

Main Manuscript:

Introduction:

11. Comment: 1. The sentence “Adhering to these vaccines for children….. deaths among children globally”. Repeated use of the same word in one sentence affects the quality. The authors should revise this sentence and similar sentences (e.g. “… led to increased globally attention which led to…” to improve the quality of the paper.

Response: The authors have addressed this comment [see page 1, line 26 and page 2, line 67-69].

12. Comment: 2. There is the need for consistency in the use of percentage (e.g. “From 47% in 1988 to 79 percent…” and “.. in 1974 where less than five percent…”

Response: The authors have addressed this comment [see page 2, line 75-77].

13. Comment: 3. The sentences “There was, however, …” and “The present study, therefore,…”should be revised.

Response: The authors have revised these sentences [see page 2, line 85-87].

Methods:

14. Comment: 1. The study site or setting is missing in the methods section. This should be included.

Response: The authors have included the study site in the methods section [see page 3, lines 93-117].

15. Comment: 2. The authors should be consistent in the use of terms: “complete vaccination” vs “full immunization coverage” vs “full immunization” vs “full childhood vaccination”,

Response: The authors have addressed this comment to take care of the inconsistencies in the entire manuscript.

16. Comment: 3. The in-text citation varies between numbers (e.g. [2,3]) and names (Armah et al., 2016). This should be revised appropriately.

Response: The authors have appropriately revised all the references in the manuscript to meet the journal’s requirements.

17. Comment: 4. This sentence is unclear “At the multivariate analysis, a Binary Logistic Regression Model was fitted”. The use of capital letters and clarity of the sentence should be review.

Response: The authors have reviewed the sentence [see page 4, line 179].

18. Comment: 5. In addition, the authors did not show the crude odds ratio before presenting their adjusted logistic regression results. This may be confusing to the readers.

Response: The authors have added the crude odds ratio to the Table 2.

19. Comment: 6. The institution that approved is the “Ghana Health Service Ethics Review Committee” not Ethical Review Committee of Ghana Health Service”.

Response: The authors have addressed this comment by changing “Ethical Review Committee of Ghana Health Service” to “Ghana Health Service Ethics Review Committee (GHS-ERC)”(see page 5, line 186).

Results:

20. Comment: 1. The sentence “From the Figure, it was seen that complete vaccination has increased over the years while income vaccination has been reducing”. This needs revision.

Response: We have revised the sentence [see page 5, line 197-198].

21. Comment: 2. The authors have used the word “respondents” inappropriately in the description of the results. (e.g. “More than half of respondents (66.2%) had mothers from rural areas”). This implies that children (12-23 months of age) were the respondents. Rather the mothers were the respondents in the current study. Thus, the entire description of table 1 should be revised.

Response: The authors have revised the results section [see page5- 6]

22. Comment: 3. There are many inaccuracies between the description of the results and figures found on tables 1 (e.g. 66.2% instead of 69.2%)

Response: The authors have revised the section to address all the inaccuracies [ see page 5-6].

23. Comment: 4. Another example, “More than 42 percent instead of 41.2%”. Thus, the presentation of the results under table 1 should be carefully done.

Response: The authors have revised the manuscript to address all the inaccuracies.

24. Comment: 5. The authors also started a couple of sentences with percentages. E.g. 75.2% of the respondents ….., 13.2% of children …, 32.4% of the respondents…”. The authors should consider reviewing the results section of the paper.

Response: The authors have reviewed and addressed all the sentences that were started with percentages.

25. Comment: 6. The authors presented some p-values as “P<0.000”. They should revise it to p<0.001.

Response: We have addressed the p-values of the manuscript (see page 11-14]

26. Comment: 7. Both the presentation of the multivariate results in table 2 and the description of the same findings need improvement. The quality of the table is low.

Response: The authors have improved the quality of the Table 2

Discussion:

27. Comment: 1. Overall, the discussion is short and unclear.

Response: The authors have addressed this comment by revising the discussion section of the manuscript [see page 7-8]. 

28. Comment: 2. The sentence “We aimed at assessing the trends and determinates ….”. The word “determinates” should be revised

Response: The authors have revised the word “determinates” to “determinants”. 

29. Comment: 3. The sentences “It was seen from the findings…from 1998 to 2014” and “That is, there was increase from 85.1% in 1998 to 95.2% in 2014”. This should be revised.

Response: The authors have revised and addressed this [see page 5].

30. Comment: 4. The sentence “…areas had a low odd of getting vaccination…” should also be revised.

Response: The authors have addressed this comment [see page 6].

31. Comment: 5. There are problems with the in-text citation under the discussion. This needs attention and revision.

Response: The authors have resolved the problems with the in-text citation under the discussion section of the manuscript[see page 6-7]

32. Comment: 6. The sentence “This finding confirms the results of a study conducted by [27, 30]”. This sentence is incomplete.

Response: The authors have addressed this comment to make this sentence complete [see page 6, line 262-264].

Limitations:

33. Comment: The authors should limit the study limitations to their study. For example, “There is the possibility of social desirability bias…” This may not be appropriate for the present study based on the methods.

Response: The limitation section has been revised [see page 8]

Abbreviations:

34. Comment: 1. Some abbreviations were not defined at first use e.g. SNNPRS.

Response: The abbreviation has been specified in the manuscript (page 7 line 269]. 

References:

35. Comment: 1. The references need formatting. For example, it is unclear to know the author of the first reference.

Response: The authors have checked and formatted the references accordingly

36. Comment: 2. The same author is presented in three different formats e.g. World Health Organization, WHO and W.H.O. The authors should use one format.

Response: This reference has been formatted accordingly to make it consistent

37. Comment: 3. There are also different in-text citations in the paper. The authors should address the inconsistencies.

Response: The authors have revised the entire manuscript to correct all the in-text citations

---

## [Decision Letter · Decision Letter 1]

28 Jul 2020

PONE-D-20-10150R1

Trend and determinants of complete vaccination coverage among children aged 12-23 months in Ghana: Analysis of data from the 1998 to 2014 Ghana Demographic and Health Surveys

PLOS ONE

Dear Dr. Budu,

Thank you for submitting your manuscript to PLOS ONE. After careful consideration, we feel that it has merit but does not fully meet PLOS ONE’s publication criteria as it currently stands. Therefore, we invite you to submit a revised version of the manuscript that addresses the points raised during the review process.

We look forward to receiving your revised manuscript.

Kind regards,

Yacob Zereyesus, Ph.D.

Academic Editor

PLOS ONE

Reviewers' comments:

Reviewer's Responses to Questions

**Comments to the Author**

1. If the authors have adequately addressed your comments raised in a previous round of review and you feel that this manuscript is now acceptable for publication, you may indicate that here to bypass the “Comments to the Author” section, enter your conflict of interest statement in the “Confidential to Editor” section, and submit your "Accept" recommendation.

Reviewer #1: All comments have been addressed

2. Is the manuscript technically sound, and do the data support the conclusions?

Reviewer #1: Yes

3. Has the statistical analysis been performed appropriately and rigorously? 

Reviewer #1: Yes

4. Have the authors made all data underlying the findings in their manuscript fully available?

Reviewer #1: Yes

5. Is the manuscript presented in an intelligible fashion and written in standard English?

Reviewer #1: Yes

6. Review Comments to the Author

Reviewer #1: Title: Trend and determinants of complete vaccination coverage among children aged 12-23 months in Ghana: Analysis of data from the 1998 to 2014 Ghana Demographic and Health Surveys

Comments to the Authors:

General Comments:

The authors have addressed the comments. However, minor issues were found for consideration.

Title:

The title is very clear. However, the word “trends” could be changed to “trend” since there is figure 1 which depicts fully immunization.

Addressed adequately

Abstract:

1. The title used the word “immunization” while the first sentence under the background of the abstract is “vaccination”. The authors should consider rephrasing either on the title or the first sentence under the background.

Adequately addressed

2. On the title, the word “trends” is used while “trend” is applied under background.

Adequately addressed

3. Inconsistency in the use of “immunization” vs “immunisation” and “full childhood vaccination” vs “full immunization coverage” and “worldwide” vs “globally”. The authors should be consistent.

Adequately addressed. However, the sentence starting with “Information about 5,119 children aged 12-23…” needs revision (Page 1, line 31).

4. The sentence “Both bivariate and multivariate analysis were….” The word “analysis” should be changed to “analyses”.

Adequately addressed

5. The logistic results (Odds Ratio and 95%CI) are inaccurate and not found in table 2. Moreover, the table2 has adjusted Odds Ratio (aOR). It is unclear whether the results under the abstract are from crude Odds Ratio (OR) or adjusted Odds Ratio (aOR).

Adequately addressed

6. The sentence “The study revealed that there has been an increase in the coverage of complete immunization …. with incomplete vaccination declining”. The second part of the sentence should be removed.

Not adequately addressed. The above sentence is still found on the conclusion section of the paper.

7. From 1998 to 2014, there was about 10.0% improvement in full immunization (85.1% to 95.2%), how did the authors determine “substantial gains”?

Adequately addressed

8. The “Ethical Review Committee of Ghana Health Service” should be changed to the “Ghana Health Service Ethics Review Committee” often written as GHSERC.

Adequately addressed

Main Manuscript:

Introduction:

1. The sentence “Adhering to these vaccines for children….. deaths among children globally”. Repeated use of the same word in one sentence affects the quality. The authors should revise this sentence and similar sentences (e.g. “… led to increased globally attention which led to…” to improve the quality of the paper.

Adequately addressed. However, the sentence starting with “According to the GDHS 2014 report, there has been in increase in vaccination coverage…” should be revised (page 2, line 75)

2. There is the need for consistency in the use of percentage (e.g. “From 47% in 1988 to 79 percent…” and “.. in 1974 where less than five percent…”

The authors have addressed it.

3. The sentences “There was, however, …” and “The present study, therefore,…”should be revised.

Adequately addressed

Methods:

1. The study site or setting is missing in the methods section. This should be included.

Adequately addressed

2. The authors should be consistent in the use of terms: “complete vaccination” vs “full immunization coverage” vs “full immunization” vs “full childhood vaccination”,

The authors have addressed it.

3. The in-text citation varies between numbers (e.g. [2,3]) and names (Armah et al., 2016). This should be revised appropriately.

Adequately addressed

4. This sentence is unclear “At the multivariate analysis, a Binary Logistic Regression Model was fitted”. The use of capital letters and clarity of the sentence should be review.

Adequately addressed. However, the word “crosstabulation” should be changed to “cross tabulation” (page 4, line 175)

5. In addition, the authors did not show the crude odds ratio before presenting their adjusted logistic regression results. This may be confusing to the readers.

Adequately addressed

6. The institution that approved is the “Ghana Health Service Ethics Review Committee” not Ethical Review Committee of Ghana Health Service”.

Adequately addressed

Results:

1. The sentence “From the Figure, it was seen that complete vaccination has increased over the years while incomplete vaccination has been reducing”. This needs revision.

Not adequately addressed. This study can be found under conclusion of the paper (page 7, line 306 and 307).

The use of socio-demographic characteristic twice in a sentence affects the quality. Please, revise “Table 1 presents the results on socio-demographic characteristics and the proportion of childhood vaccination across socio-demographic characteristics” (Page 5, line 207 and 208).

2. The authors have used the word “respondents” inappropriately in the description of the results. (e.g. “More than half of respondents (66.2%) had mothers from rural areas”). This implies that children (12-23 months of age) were the respondents. Rather the mothers were the respondents in the current study. Thus, the entire description of table 1 should be revised.

Adequately addressed

3. There are many inaccuracies between the description of the results and figures found on tables 1 (e.g. 66.2% instead of 69.2%)

Adequately addressed

4. Another example, “More than 42 percent instead of 41.2%”. Thus, the presentation of the results under table 1 should be carefully done.

Adequately addressed

5. The authors also started a couple of sentences with percentages. E.g. 75.2% of the respondents ….., 13.2% of children …, 32.4% of the respondents…”. The authors should consider reviewing the results section of the paper.

Adequately addressed

6. The authors presented some p-values as “P<0.000”. They should revise it to p<0.001.

Adequately addressed

7. Both the presentation of the multivariate results in table 2 and the description of the same findings need improvement. The quality of the table is low.

Adequately addressed

Discussion:

1. Overall, the discussion is short and unclear.

Adequately addressed

2. The sentence “We aimed at assessing the trends and determinates ….”. The word “determinates” should be revised

Adequately addressed

3. The sentences “It was seen from the findings…from 1998 to 2014” and “That is, there was increase from 85.1% in 1998 to 95.2% in 2014”. This should be revised.

Adequately addressed

4. The sentence “…areas had a low odd of getting vaccination…” should also be revised.

Adequately addressed

5. There are problems with the in-text citation under the discussion. This needs attention and revision.

Adequately addressed

6. The sentence “This finding confirms the results of a study conducted by [27, 30]”. This sentence is incomplete.

Adequately addressed

Limitations:

The authors should limit the study limitations to their study. For example, “There is the possibility of social desirability bias…” This may not be appropriate for the present study based on the methods.

Adequately addressed

Abbreviations:

1. Some abbreviations were not defined at first use e.g. SNNPRS.

Adequately addressed

References:

1. The references need formatting. For example, it is unclear to know the author of the first reference.

2. The same author is presented in three different formats e.g. World Health Organization, WHO and W.H.O. The authors should use one format.

3. There are also different in-text citations in the paper. The authors should address the inconsistencies.

Adequately addressed

7. PLOS authors have the option to publish the peer review history of their article (what does this mean?). If published, this will include your full peer review and any attached files.

Reviewer #1: No

---

## [Author Response · Author response to Decision Letter 1]

28 Jul 2020

AUTHORS’ RESPONSE TO REVIEWS

Title: Trend and determinants of complete vaccination coverage among children aged 12-23 months in Ghana: Analysis of data from the 1998 to 2014 Ghana Demographic and Health Surveys

Dear Editor and Reviewer (s),

On behalf of all authors, I convey our gratitude to you for the critical and constructive review that has led to the improvement of our paper entitled “Trend and determinants of complete vaccination coverage among children aged 12-23 months in Ghana: Analysis of data from the 1998 to 2014 Ghana Demographic and Health Surveys”. We have revised the manuscript based on the comments raised. In the following detailed response, we address each comment calling for changes point-by-point, indicating where relevant additional texts have been added to the body of the manuscript. Most of the changes have been indicated in yellow colour. We believe the manuscript has improved substantively and will be published in your reputable journal, PLOS ONE 

Version:2

Manuscript ID: PONE-D-20-10150_R1

Date: 28/07/2020

Reviewer #1: Title: Trend and determinants of complete vaccination coverage among children aged 12-23 months in Ghana: Analysis of data from the 1998 to 2014 Ghana Demographic and Health Surveys

Comments to the Authors:

General Comments:

The authors have addressed the comments. However, minor issues were found for consideration.

Title:

The title is very clear. However, the word “trends” could be changed to “trend” since there is figure 1 which depicts fully immunization.

Addressed adequately

Abstract:

1. The title used the word “immunization” while the first sentence under the background of the abstract is “vaccination”. The authors should consider rephrasing either on the title or the first sentence under the background.

Adequately addressed

2. On the title, the word “trends” is used while “trend” is applied under background.

Adequately addressed

3. Inconsistency in the use of “immunization” vs “immunisation” and “full childhood vaccination” vs “full immunization coverage” and “worldwide” vs “globally”. The authors should be consistent.

Adequately addressed. However, the sentence starting with “Information about 5,119 children aged 12-23…” needs revision (Page 1, line 31).

Response: This has been revised. See page 1 line 31-32

4. The sentence “Both bivariate and multivariate analysis were….” The word “analysis” should be changed to “analyses”.

Adequately addressed

5. The logistic results (Odds Ratio and 95%CI) are inaccurate and not found in table 2. Moreover, the table2 has adjusted Odds Ratio (aOR). It is unclear whether the results under the abstract are from crude Odds Ratio (OR) or adjusted Odds Ratio (aOR).

Adequately addressed

6. The sentence “The study revealed that there has been an increase in the coverage of complete immunization …. with incomplete vaccination declining”. The second part of the sentence should be removed.

Not adequately addressed. The above sentence is still found on the conclusion section of the paper.

Response: This has been addressed. See page 7 line 307-308. 

7. From 1998 to 2014, there was about 10.0% improvement in full immunization (85.1% to 95.2%), how did the authors determine “substantial gains”?

Adequately addressed

8. The “Ethical Review Committee of Ghana Health Service” should be changed to the “Ghana Health Service Ethics Review Committee” often written as GHSERC.

Adequately addressed

Main Manuscript:

Introduction:

1. The sentence “Adhering to these vaccines for children….. deaths among children globally”. Repeated use of the same word in one sentence affects the quality. The authors should revise this sentence and similar sentences (e.g. “… led to increased globally attention which led to…” to improve the quality of the paper.

Adequately addressed. However, the sentence starting with “According to the GDHS 2014 report, there has been in increase in vaccination coverage…” should be revised (page 2, line 75)

Response: This has been addressed. See Page 2 line 75-76. 

2. There is the need for consistency in the use of percentage (e.g. “From 47% in 1988 to 79 percent…” and “.. in 1974 where less than five percent…”

The authors have addressed it.

3. The sentences “There was, however, …” and “The present study, therefore,…”should be revised.

Adequately addressed

Methods:

1. The study site or setting is missing in the methods section. This should be included.

Adequately addressed

2. The authors should be consistent in the use of terms: “complete vaccination” vs “full immunization coverage” vs “full immunization” vs “full childhood vaccination”,

The authors have addressed it.

3. The in-text citation varies between numbers (e.g. [2,3]) and names (Armah et al., 2016). This should be revised appropriately.

Adequately addressed

4. This sentence is unclear “At the multivariate analysis, a Binary Logistic Regression Model was fitted”. The use of capital letters and clarity of the sentence should be review.

Adequately addressed. However, the word “crosstabulation” should be changed to “cross tabulation” (page 4, line 175)

Response: This has been revised. See Page 4 line 175.

5. In addition, the authors did not show the crude odds ratio before presenting their adjusted logistic regression results. This may be confusing to the readers.

Adequately addressed

6. The institution that approved is the “Ghana Health Service Ethics Review Committee” not Ethical Review Committee of Ghana Health Service”.

Adequately addressed

Results:

1. The sentence “From the Figure, it was seen that complete vaccination has increased over the years while incomplete vaccination has been reducing”. This needs revision.

Not adequately addressed. This study can be found under conclusion of the paper (page 7, line 306 and 307).

Response: This has been revised. See Page 7 line 306-307. 

The use of socio-demographic characteristic twice in a sentence affects the quality. Please, revise “Table 1 presents the results on socio-demographic characteristics and the proportion of childhood vaccination across socio-demographic characteristics” (Page 5, line 207 and 208).

2. The authors have used the word “respondents” inappropriately in the description of the results. (e.g. “More than half of respondents (66.2%) had mothers from rural areas”). This implies that children (12-23 months of age) were the respondents. Rather the mothers were the respondents in the current study. Thus, the entire description of table 1 should be revised.

Adequately addressed

3. There are many inaccuracies between the description of the results and figures found on tables 1 (e.g. 66.2% instead of 69.2%)

Adequately addressed

4. Another example, “More than 42 percent instead of 41.2%”. Thus, the presentation of the results under table 1 should be carefully done.

Adequately addressed

5. The authors also started a couple of sentences with percentages. E.g. 75.2% of the respondents ….., 13.2% of children …, 32.4% of the respondents…”. The authors should consider reviewing the results section of the paper.

Adequately addressed

6. The authors presented some p-values as “P<0.000”. They should revise it to p<0.001.

Adequately addressed

7. Both the presentation of the multivariate results in table 2 and the description of the same findings need improvement. The quality of the table is low.

Adequately addressed

Discussion:

1. Overall, the discussion is short and unclear.

Adequately addressed

2. The sentence “We aimed at assessing the trends and determinates ….”. The word “determinates” should be revised

Adequately addressed

3. The sentences “It was seen from the findings…from 1998 to 2014” and “That is, there was increase from 85.1% in 1998 to 95.2% in 2014”. This should be revised.

Adequately addressed

4. The sentence “…areas had a low odd of getting vaccination…” should also be revised.

Adequately addressed

5. There are problems with the in-text citation under the discussion. This needs attention and revision.

Adequately addressed

6. The sentence “This finding confirms the results of a study conducted by [27, 30]”. This sentence is incomplete.

Adequately addressed

Limitations:

The authors should limit the study limitations to their study. For example, “There is the possibility of social desirability bias…” This may not be appropriate for the present study based on the methods.

Adequately addressed

Abbreviations:

1. Some abbreviations were not defined at first use e.g. SNNPRS.

Adequately addressed

References:

1. The references need formatting. For example, it is unclear to know the author of the first reference.

2. The same author is presented in three different formats e.g. World Health Organization, WHO and W.H.O. The authors should use one format.

3. There are also different in-text citations in the paper. The authors should address the inconsistencies.

Adequately addressed

AUTHORS’ RESPONSE TO REVIEWS

Title: Trend and determinants of complete vaccination coverage among children aged 12-23 months in Ghana: Analysis of data from the 1998 to 2014 Ghana Demographic and Health Surveys

Dear Editor and Reviewer (s),

On behalf of all authors, I convey our gratitude to you for the critical and constructive review that has led to the improvement of our paper entitled “Trend and determinants of complete vaccination coverage among children aged 12-23 months in Ghana: Analysis of data from the 1998 to 2014 Ghana Demographic and Health Surveys”. We have revised the manuscript based on the comments raised. In the following detailed response, we address each comment calling for changes point-by-point, indicating where relevant additional texts have been added to the body of the manuscript. Most of the changes have been indicated in yellow colour. We believe the manuscript has improved substantively and will be published in your reputable journal, PLOS ONE 

Version:2

Manuscript ID: PONE-D-20-10150_R1

Date: 28/07/2020

Reviewer #1: Title: Trend and determinants of complete vaccination coverage among children aged 12-23 months in Ghana: Analysis of data from the 1998 to 2014 Ghana Demographic and Health Surveys

Comments to the Authors:

General Comments:

The authors have addressed the comments. However, minor issues were found for consideration.

Title:

The title is very clear. However, the word “trends” could be changed to “trend” since there is figure 1 which depicts fully immunization.

Addressed adequately

Abstract:

1. The title used the word “immunization” while the first sentence under the background of the abstract is “vaccination”. The authors should consider rephrasing either on the title or the first sentence under the background.

Adequately addressed

2. On the title, the word “trends” is used while “trend” is applied under background.

Adequately addressed

3. Inconsistency in the use of “immunization” vs “immunisation” and “full childhood vaccination” vs “full immunization coverage” and “worldwide” vs “globally”. The authors should be consistent.

Adequately addressed. However, the sentence starting with “Information about 5,119 children aged 12-23…” needs revision (Page 1, line 31).

Response: This has been revised. See page 1 line 31-32

4. The sentence “Both bivariate and multivariate analysis were….” The word “analysis” should be changed to “analyses”.

Adequately addressed

5. The logistic results (Odds Ratio and 95%CI) are inaccurate and not found in table 2. Moreover, the table2 has adjusted Odds Ratio (aOR). It is unclear whether the results under the abstract are from crude Odds Ratio (OR) or adjusted Odds Ratio (aOR).

Adequately addressed

6. The sentence “The study revealed that there has been an increase in the coverage of complete immunization …. with incomplete vaccination declining”. The second part of the sentence should be removed.

Not adequately addressed. The above sentence is still found on the conclusion section of the paper.

Response: This has been addressed. See page 7 line 307-308. 

7. From 1998 to 2014, there was about 10.0% improvement in full immunization (85.1% to 95.2%), how did the authors determine “substantial gains”?

Adequately addressed

8. The “Ethical Review Committee of Ghana Health Service” should be changed to the “Ghana Health Service Ethics Review Committee” often written as GHSERC.

Adequately addressed

Main Manuscript:

Introduction:

1. The sentence “Adhering to these vaccines for children….. deaths among children globally”. Repeated use of the same word in one sentence affects the quality. The authors should revise this sentence and similar sentences (e.g. “… led to increased globally attention which led to…” to improve the quality of the paper.

Adequately addressed. However, the sentence starting with “According to the GDHS 2014 report, there has been in increase in vaccination coverage…” should be revised (page 2, line 75)

Response: This has been addressed. See Page 2 line 75-76. 

2. There is the need for consistency in the use of percentage (e.g. “From 47% in 1988 to 79 percent…” and “.. in 1974 where less than five percent…”

The authors have addressed it.

3. The sentences “There was, however, …” and “The present study, therefore,…”should be revised.

Adequately addressed

Methods:

1. The study site or setting is missing in the methods section. This should be included.

Adequately addressed

2. The authors should be consistent in the use of terms: “complete vaccination” vs “full immunization coverage” vs “full immunization” vs “full childhood vaccination”,

The authors have addressed it.

3. The in-text citation varies between numbers (e.g. [2,3]) and names (Armah et al., 2016). This should be revised appropriately.

Adequately addressed

4. This sentence is unclear “At the multivariate analysis, a Binary Logistic Regression Model was fitted”. The use of capital letters and clarity of the sentence should be review.

Adequately addressed. However, the word “crosstabulation” should be changed to “cross tabulation” (page 4, line 175)

Response: This has been revised. See Page 4 line 175.

5. In addition, the authors did not show the crude odds ratio before presenting their adjusted logistic regression results. This may be confusing to the readers.

Adequately addressed

6. The institution that approved is the “Ghana Health Service Ethics Review Committee” not Ethical Review Committee of Ghana Health Service”.

Adequately addressed

Results:

1. The sentence “From the Figure, it was seen that complete vaccination has increased over the years while incomplete vaccination has been reducing”. This needs revision.

Not adequately addressed. This study can be found under conclusion of the paper (page 7, line 306 and 307).

Response: This has been revised. See Page 7 line 306-307. 

The use of socio-demographic characteristic twice in a sentence affects the quality. Please, revise “Table 1 presents the results on socio-demographic characteristics and the proportion of childhood vaccination across socio-demographic characteristics” (Page 5, line 207 and 208).

2. The authors have used the word “respondents” inappropriately in the description of the results. (e.g. “More than half of respondents (66.2%) had mothers from rural areas”). This implies that children (12-23 months of age) were the respondents. Rather the mothers were the respondents in the current study. Thus, the entire description of table 1 should be revised.

Adequately addressed

3. There are many inaccuracies between the description of the results and figures found on tables 1 (e.g. 66.2% instead of 69.2%)

Adequately addressed

4. Another example, “More than 42 percent instead of 41.2%”. Thus, the presentation of the results under table 1 should be carefully done.

Adequately addressed

5. The authors also started a couple of sentences with percentages. E.g. 75.2% of the respondents ….., 13.2% of children …, 32.4% of the respondents…”. The authors should consider reviewing the results section of the paper.

Adequately addressed

6. The authors presented some p-values as “P<0.000”. They should revise it to p<0.001.

Adequately addressed

7. Both the presentation of the multivariate results in table 2 and the description of the same findings need improvement. The quality of the table is low.

Adequately addressed

Discussion:

1. Overall, the discussion is short and unclear.

Adequately addressed

2. The sentence “We aimed at assessing the trends and determinates ….”. The word “determinates” should be revised

Adequately addressed

3. The sentences “It was seen from the findings…from 1998 to 2014” and “That is, there was increase from 85.1% in 1998 to 95.2% in 2014”. This should be revised.

Adequately addressed

4. The sentence “…areas had a low odd of getting vaccination…” should also be revised.

Adequately addressed

5. There are problems with the in-text citation under the discussion. This needs attention and revision.

Adequately addressed

6. The sentence “This finding confirms the results of a study conducted by [27, 30]”. This sentence is incomplete.

Adequately addressed

Limitations:

The authors should limit the study limitations to their study. For example, “There is the possibility of social desirability bias…” This may not be appropriate for the present study based on the methods.

Adequately addressed

Abbreviations:

1. Some abbreviations were not defined at first use e.g. SNNPRS.

Adequately addressed

References:

1. The references need formatting. For example, it is unclear to know the author of the first reference.

2. The same author is presented in three different formats e.g. World Health Organization, WHO and W.H.O. The authors should use one format.

3. There are also different in-text citations in the paper. The authors should address the inconsistencies.

Adequately addressed

---

## [Editor Report · Decision Letter 2]

25 Aug 2020

PONE-D-20-10150R2

Trend and determinants of complete vaccination coverage among children aged 12-23 months in Ghana: Analysis of data from the 1998 to 2014 Ghana Demographic and Health Surveys

PLOS ONE

Dear Dr. Budu,

Thank you for submitting your manuscript to PLOS ONE. After careful consideration, we feel that it has merit but does not fully meet PLOS ONE’s publication criteria as it currently stands. Therefore, we invite you to submit a revised version of the manuscript that addresses the points raised during the review process.

We look forward to receiving your revised manuscript.

Kind regards,

Yacob Zereyesus, Ph.D.

Academic Editor

PLOS ONE

Additional Editor Comments (if provided):

Title: Trend and determinants of complete vaccination coverage among children aged 12-23 months in Ghana: Analysis of data from the 1998 to 2014 Ghana Demographic and Health Surveys

Comments to the Authors:

General Comments:

The authors have addressed the comments. However, minor issues were found for consideration.

Title:

The title is very clear. However, the word “trends” could be changed to “trend” since there is figure 1 which depicts fully immunization.

Addressed adequately

Abstract:

1. The title used the word “immunization” while the first sentence under the background of the abstract is “vaccination”. The authors should consider rephrasing either on the title or the first sentence under the background.

Adequately addressed

2. On the title, the word “trends” is used while “trend” is applied under background.

Adequately addressed

3. Inconsistency in the use of “immunization” vs “immunisation” and “full childhood vaccination” vs “full immunization coverage” and “worldwide” vs “globally”. The authors should be consistent.

Adequately addressed. However, the sentence starting with “Information about 5,119 children aged 12-23…” needs revision (Page 1, line 31).

4. The sentence “Both bivariate and multivariate analysis were….” The word “analysis” should be changed to “analyses”.

Adequately addressed

5. The logistic results (Odds Ratio and 95%CI) are inaccurate and not found in table 2. Moreover, the table2 has adjusted Odds Ratio (aOR). It is unclear whether the results under the abstract are from crude Odds Ratio (OR) or adjusted Odds Ratio (aOR).

Adequately addressed

6. The sentence “The study revealed that there has been an increase in the coverage of complete immunization …. with incomplete vaccination declining”. The second part of the sentence should be removed.

Not adequately addressed. The above sentence is still found on the conclusion section of the paper.

7. From 1998 to 2014, there was about 10.0% improvement in full immunization (85.1% to 95.2%), how did the authors determine “substantial gains”?

Adequately addressed

8. The “Ethical Review Committee of Ghana Health Service” should be changed to the “Ghana Health Service Ethics Review Committee” often written as GHSERC.

Adequately addressed

Main Manuscript:

Introduction:

1. The sentence “Adhering to these vaccines for children….. deaths among children globally”. Repeated use of the same word in one sentence affects the quality. The authors should revise this sentence and similar sentences (e.g. “… led to increased globally attention which led to…” to improve the quality of the paper.

Adequately addressed. However, the sentence starting with “According to the GDHS 2014 report, there has been in increase in vaccination coverage…” should be revised (page 2, line 75)

2. There is the need for consistency in the use of percentage (e.g. “From 47% in 1988 to 79 percent…” and “.. in 1974 where less than five percent…”

The authors have addressed it.

3. The sentences “There was, however, …” and “The present study, therefore,…”should be revised.

Adequately addressed

Methods:

1. The study site or setting is missing in the methods section. This should be included.

Adequately addressed

2. The authors should be consistent in the use of terms: “complete vaccination” vs “full immunization coverage” vs “full immunization” vs “full childhood vaccination”,

The authors have addressed it.

3. The in-text citation varies between numbers (e.g. [2,3]) and names (Armah et al., 2016). This should be revised appropriately.

Adequately addressed

4. This sentence is unclear “At the multivariate analysis, a Binary Logistic Regression Model was fitted”. The use of capital letters and clarity of the sentence should be review.

Adequately addressed. However, the word “crosstabulation” should be changed to “cross tabulation” (page 4, line 175)

5. In addition, the authors did not show the crude odds ratio before presenting their adjusted logistic regression results. This may be confusing to the readers.

Adequately addressed

6. The institution that approved is the “Ghana Health Service Ethics Review Committee” not Ethical Review Committee of Ghana Health Service”.

Adequately addressed

Results:

1. The sentence “From the Figure, it was seen that complete vaccination has increased over the years while incomplete vaccination has been reducing”. This needs revision.

Not adequately addressed. This study can be found under conclusion of the paper (page 7, line 306 and 307).

The use of socio-demographic characteristic twice in a sentence affects the quality. Please, revise “Table 1 presents the results on socio-demographic characteristics and the proportion of childhood vaccination across socio-demographic characteristics” (Page 5, line 207 and 208).

2. The authors have used the word “respondents” inappropriately in the description of the results. (e.g. “More than half of respondents (66.2%) had mothers from rural areas”). This implies that children (12-23 months of age) were the respondents. Rather the mothers were the respondents in the current study. Thus, the entire description of table 1 should be revised.

Adequately addressed

3. There are many inaccuracies between the description of the results and figures found on tables 1 (e.g. 66.2% instead of 69.2%)

Adequately addressed

4. Another example, “More than 42 percent instead of 41.2%”. Thus, the presentation of the results under table 1 should be carefully done.

Adequately addressed

5. The authors also started a couple of sentences with percentages. E.g. 75.2% of the respondents ….., 13.2% of children …, 32.4% of the respondents…”. The authors should consider reviewing the results section of the paper.

Adequately addressed

6. The authors presented some p-values as “P<0.000”. They should revise it to p<0.001.

Adequately addressed

7. Both the presentation of the multivariate results in table 2 and the description of the same findings need improvement. The quality of the table is low.

Adequately addressed

Discussion:

1. Overall, the discussion is short and unclear.

Adequately addressed

2. The sentence “We aimed at assessing the trends and determinates ….”. The word “determinates” should be revised

Adequately addressed

3. The sentences “It was seen from the findings…from 1998 to 2014” and “That is, there was increase from 85.1% in 1998 to 95.2% in 2014”. This should be revised.

Adequately addressed

4. The sentence “…areas had a low odd of getting vaccination…” should also be revised.

Adequately addressed

5. There are problems with the in-text citation under the discussion. This needs attention and revision.

Adequately addressed

6. The sentence “This finding confirms the results of a study conducted by [27, 30]”. This sentence is incomplete.

Adequately addressed

Limitations:

The authors should limit the study limitations to their study. For example, “There is the possibility of social desirability bias…” This may not be appropriate for the present study based on the methods.

Adequately addressed

Abbreviations:

1. Some abbreviations were not defined at first use e.g. SNNPRS.

Adequately addressed

References:

1. The references need formatting. For example, it is unclear to know the author of the first reference.

2. The same author is presented in three different formats e.g. World Health Organization, WHO and W.H.O. The authors should use one format.

3. There are also different in-text citations in the paper. The authors should address the inconsistencies.

Adequately addressed

---

## [Author Response · Author response to Decision Letter 2]

4 Sep 2020

AUTHORS’ RESPONSE TO REVIEWS

Title: Trend and determinants of complete vaccination coverage among children aged 12-23 months in Ghana: Analysis of data from the 1998 to 2014 Ghana Demographic and Health Surveys

Dear Editor and Reviewer (s),

On behalf of all authors, I convey our gratitude to you for the critical and constructive review that has led to the improvement of our paper entitled “Trend and determinants of complete vaccination coverage among children aged 12-23 months in Ghana: Analysis of data from the 1998 to 2014 Ghana Demographic and Health Surveys”. We have revised the manuscript based on the comments raised. In the following detailed response, we address each comment calling for changes point-by-point, indicating where relevant additional texts have been added to the body of the manuscript. Most of the changes have been indicated in yellow colour. We believe the manuscript has improved substantively and will be published in your reputable journal, PLOS ONE 

Version:3

Manuscript ID: PONE-D-20-10150_R2

Date: 28/07/2020

Reviewer #1: Title: Trend and determinants of complete vaccination coverage among children aged 12-23 months in Ghana: Analysis of data from the 1998 to 2014 Ghana Demographic and Health Surveys

Comments to the Authors:

General Comments:

The authors have addressed the comments. However, minor issues were found for consideration.

Title:

The title is very clear. However, the word “trends” could be changed to “trend” since there is figure 1 which depicts fully immunization.

Addressed adequately

Abstract:

1. The title used the word “immunization” while the first sentence under the background of the abstract is “vaccination”. The authors should consider rephrasing either on the title or the first sentence under the background.

Adequately addressed

2. On the title, the word “trends” is used while “trend” is applied under background.

Adequately addressed

3. Inconsistency in the use of “immunization” vs “immunisation” and “full childhood vaccination” vs “full immunization coverage” and “worldwide” vs “globally”. The authors should be consistent.

Adequately addressed. However, the sentence starting with “Information about 5,119 children aged 12-23…” needs revision (Page 1, line 31).

Response: This has been revised. See page 1 line 31-32

4. The sentence “Both bivariate and multivariate analysis were….” The word “analysis” should be changed to “analyses”.

Adequately addressed

5. The logistic results (Odds Ratio and 95%CI) are inaccurate and not found in table 2. Moreover, the table2 has adjusted Odds Ratio (aOR). It is unclear whether the results under the abstract are from crude Odds Ratio (OR) or adjusted Odds Ratio (aOR).

Adequately addressed

6. The sentence “The study revealed that there has been an increase in the coverage of complete immunization …. with incomplete vaccination declining”. The second part of the sentence should be removed.

Not adequately addressed. The above sentence is still found on the conclusion section of the paper.

Response: This has been addressed. See page 7 line 307-308. 

7. From 1998 to 2014, there was about 10.0% improvement in full immunization (85.1% to 95.2%), how did the authors determine “substantial gains”?

Adequately addressed

8. The “Ethical Review Committee of Ghana Health Service” should be changed to the “Ghana Health Service Ethics Review Committee” often written as GHSERC.

Adequately addressed

Main Manuscript:

Introduction:

1. The sentence “Adhering to these vaccines for children….. deaths among children globally”. Repeated use of the same word in one sentence affects the quality. The authors should revise this sentence and similar sentences (e.g. “… led to increased globally attention which led to…” to improve the quality of the paper.

Adequately addressed. However, the sentence starting with “According to the GDHS 2014 report, there has been in increase in vaccination coverage…” should be revised (page 2, line 75)

Response: This has been addressed. See Page 2 line 75-76. 

2. There is the need for consistency in the use of percentage (e.g. “From 47% in 1988 to 79 percent…” and “.. in 1974 where less than five percent…”

The authors have addressed it.

3. The sentences “There was, however, …” and “The present study, therefore,…”should be revised.

Adequately addressed

Methods:

1. The study site or setting is missing in the methods section. This should be included.

Adequately addressed

2. The authors should be consistent in the use of terms: “complete vaccination” vs “full immunization coverage” vs “full immunization” vs “full childhood vaccination”,

The authors have addressed it.

3. The in-text citation varies between numbers (e.g. [2,3]) and names (Armah et al., 2016). This should be revised appropriately.

Adequately addressed

4. This sentence is unclear “At the multivariate analysis, a Binary Logistic Regression Model was fitted”. The use of capital letters and clarity of the sentence should be review.

Adequately addressed. However, the word “crosstabulation” should be changed to “cross tabulation” (page 4, line 175)

Response: This has been revised. See Page 4 line 175.

5. In addition, the authors did not show the crude odds ratio before presenting their adjusted logistic regression results. This may be confusing to the readers.

Adequately addressed

6. The institution that approved is the “Ghana Health Service Ethics Review Committee” not Ethical Review Committee of Ghana Health Service”.

Adequately addressed

Results:

1. The sentence “From the Figure, it was seen that complete vaccination has increased over the years while incomplete vaccination has been reducing”. This needs revision.

Not adequately addressed. This study can be found under conclusion of the paper (page 7, line 306 and 307).

Response: This has been revised. See Page 7 line 306-307. 

The use of socio-demographic characteristic twice in a sentence affects the quality. Please, revise “Table 1 presents the results on socio-demographic characteristics and the proportion of childhood vaccination across socio-demographic characteristics” (Page 5, line 207 and 208).

2. The authors have used the word “respondents” inappropriately in the description of the results. (e.g. “More than half of respondents (66.2%) had mothers from rural areas”). This implies that children (12-23 months of age) were the respondents. Rather the mothers were the respondents in the current study. Thus, the entire description of table 1 should be revised.

Adequately addressed

3. There are many inaccuracies between the description of the results and figures found on tables 1 (e.g. 66.2% instead of 69.2%)

Adequately addressed

4. Another example, “More than 42 percent instead of 41.2%”. Thus, the presentation of the results under table 1 should be carefully done.

Adequately addressed

5. The authors also started a couple of sentences with percentages. E.g. 75.2% of the respondents ….., 13.2% of children …, 32.4% of the respondents…”. The authors should consider reviewing the results section of the paper.

Adequately addressed

6. The authors presented some p-values as “P<0.000”. They should revise it to p<0.001.

Adequately addressed

7. Both the presentation of the multivariate results in table 2 and the description of the same findings need improvement. The quality of the table is low.

Adequately addressed

Discussion:

1. Overall, the discussion is short and unclear.

Adequately addressed

2. The sentence “We aimed at assessing the trends and determinates ….”. The word “determinates” should be revised

Adequately addressed

3. The sentences “It was seen from the findings…from 1998 to 2014” and “That is, there was increase from 85.1% in 1998 to 95.2% in 2014”. This should be revised.

Adequately addressed

4. The sentence “…areas had a low odd of getting vaccination…” should also be revised.

Adequately addressed

5. There are problems with the in-text citation under the discussion. This needs attention and revision.

Adequately addressed

6. The sentence “This finding confirms the results of a study conducted by [27, 30]”. This sentence is incomplete.

Adequately addressed

Limitations:

The authors should limit the study limitations to their study. For example, “There is the possibility of social desirability bias…” This may not be appropriate for the present study based on the methods.

Adequately addressed

Abbreviations:

1. Some abbreviations were not defined at first use e.g. SNNPRS.

Adequately addressed

References:

1. The references need formatting. For example, it is unclear to know the author of the first reference.

2. The same author is presented in three different formats e.g. World Health Organization, WHO and W.H.O. The authors should use one format.

3. There are also different in-text citations in the paper. The authors should address the inconsistencies.

Adequately addressed

---

## [Editor Report · Decision Letter 3]

14 Sep 2020

Trend and determinants of complete vaccination coverage among children aged 12-23 months in Ghana: Analysis of data from the 1998 to 2014 Ghana Demographic and Health Surveys

PONE-D-20-10150R3

Dear Dr. Budu,

We’re pleased to inform you that your manuscript has been judged scientifically suitable for publication and will be formally accepted for publication once it meets all outstanding technical requirements.

Kind regards,

Yacob Zereyesus, Ph.D.

Academic Editor

PLOS ONE
---

## [Editor Report · Acceptance letter]

17 Sep 2020

PONE-D-20-10150R3 

Trend and determinants of complete vaccination coverage among children aged 12-23 months in Ghana: Analysis of data from the 1998 to 2014 Ghana Demographic and Health Surveys 

Dear Dr. Budu:

I'm pleased to inform you that your manuscript has been deemed suitable for publication in PLOS ONE. Congratulations! Your manuscript is now with our production department. 

Kind regards, 

on behalf of

Dr. Yacob Zereyesus 

Academic Editor

PLOS ONE